# The Cognitive Profile of People with High-Functioning Autism Spectrum Disorders

**DOI:** 10.3390/bs9020020

**Published:** 2019-02-20

**Authors:** Atusa Rabiee, Sayyed Ali Samadi, Behnoosh Vasaghi-Gharamaleki, Soode Hosseini, Saba Seyedin, Mohammadreza Keyhani, Ameneh Mahmoodizadeh, Fatemeh Ranjbar Kermani

**Affiliations:** 1Department of Speech and Language Pathology, School of Rehabilitation Sciences, Iran University of Medical Sciences, Tehran 15459-13487, Iran; 2Institute of Nursing and Health Research, University of Ulster, Northern Ireland BT37 0QB, UK; s.samadi@ulster.ac.uk; 3Department of Basic Sciences, School of Rehabilitation Sciences, Iran University of Medical Sciences, Tehran 15459-13487, Iran; vasaghi.b@iums.ac.ir (B.V.-G.); more_keyhani@yahoo.com (M.K.); 4Department of Psychology, School of Education and Psychology, Alzahra University, Tehran 1993893973, Iran; hoseini.soodeh@gmail.com; 5Department of Speech and Language Pathology, School of Rehabilitation Sciences, Tehran University of Medical Sciences, Tehran 11489-65111, Iran; s.seyedin86@gmail.com; 6Division of Diagnosis and Prevention, Iranian Special Education Organization, Tehran 1416935684, Iran; nmahmoudizadeh@yahoo.com; 7Autism Rehabilitation Center, Roozbeh Hospital, Tehran University of Medical Sciences, Tehran 1514945311, Iran; dr.ranjbar135@yahoo.com

**Keywords:** autism spectrum disorders, ASD, intelligence, WISC-IV, ADHD, Cognition

## Abstract

Several studies have examined the cognitive profile of people with high-functioning autism spectrum disorders (ASD) (IQ > 70), and its relationship with the symptoms of ASD and attention deficit hyperactivity disorder (ADHD), using the Wechsler Intelligence Scale for Children-IV (WISC-IV). However, no data exist on the similarities or differences in this profile in less affluent countries. The present study examined the cognitive profile and its relationship with the symptoms of ASD and ADHD in 30 subjects aged 6–16 years with high-functioning ASD and compared the results with those of 30 typically developing (TD) subjects. In line with previous research findings, the WISC-IV cognitive profile analysis of subjects with high-functioning ASD showed a good competence in Matrix Reasoning and weaknesses in Comprehension, but the main distinguishing point was the competence in processing speed in both groups. In the present study, the Verbal Comprehension Index correlated negatively with the communication symptoms, and the Working Memory Index correlated positively with the social symptoms in the ASD group. Given the similarities that exist between the results of the present research and previous studies, it may be concluded that there are similarities in the cognitive profile of individuals with ASD.

## 1. Introduction

The Diagnostic and Statistical Manual of Mental Disorders, Fifth Edition (DSM-5) defines autism spectrum disorders (ASD) as neurodevelopmental disorders [1]. ASD is observed in different cultures [2]. In Iran, an autism prevalence rate of 95.2 per 10,000 has been reported [3]. ASD can be accompanied by intellectual disability. Baio et al. [4] reported that 31% of people with ASD have an intellectual disability (full-scale IQ, FSIQ < 70). Individuals with ASD without intellectual disability (FSIQ > 70) are typically referred to as having high-functioning ASD [5]. Intelligence tests can provide interesting information about the cognitive strengths and weaknesses of people [6]. The Wechsler Intelligence Scale for Children (WISC) is one of the most commonly used tests for measuring intelligence in ASD individuals, and there is a wealth of literature on this tool. The test was updated in 2003, and its fourth edition (WISC-IV) is available with some changes made to the previous version. The WISC-IV has four indices, including the Verbal Comprehension Index (VCI), the Perceptual Reasoning Index (PRI), the Working Memory Index (WMI), and the Processing Speed Index (PSI). It also contains 10 core subtests, including Similarities, Vocabulary, Comprehension, Block Design, Picture Concepts, Matrix Reasoning, Digit Span, Letter-Number Sequencing, Coding, and Symbol Search, and five complementary subtests, including Information, Word Reasoning, Picture Completion, Arithmetic, and Cancellation [7]. One of the main points about intelligence tests, especially the WISC-IV, is that they provide valuable information about a subject’s cognitive strengths and weaknesses [6].

The first study of the cognitive profile of people with ASD using the WISC-IV was performed by Wechsler in the WISC-IV manual on 27 individuals aged 9 to 15 who had ASD without comorbid intellectual disability, and 19 ASD people aged 7 to 16, His results revealed an average cognitive profile in the individuals with ASD without comorbid intellectual disability and a low average profile in the ASD group. The PSI was the weakest index, and Coding and Symbol Search were the weakest subtests. The Comprehension subtest was also one of the weakest in the ASD group. The VCI was higher than the other indices in the individuals with ASD without comorbid intellectual disability, and the PRI was the highest in the ASD group [7]. Mayes and Calhoun [8] studied 54 children with ASD, aged 6 to 14, with high-functioning autism (HFA), and revealed the lowest scores on the WMI and PSI indices. Among the nonverbal subtests, Matrix Reasoning and Picture Concepts received the highest scores, and the subtests Coding, Symbol Search, Letter-Number Sequencing, and Digit Span received significantly lower scores than the norms specified for the tests. Oliveras-Rentas et al. [9]) conducted the WISC-IV test on 51 ASD people and found that the PSI was the most significantly low index, and the PRI had significantly higher scores than the WMI. Matrix Reasoning and Similarities were the strongest subtests, and Comprehension, Coding, and Symbol Search were the weakest. Nader, Jelenic, and Soulieres [10] examined the Wechsler cognitive profiles of 51 ASD people, 15 individuals with ASD without comorbid intellectual disability, and 42 typical development (TD) people. Their results showed that, in the ASD group, the PRI was higher than the full-scale IQ (FSIQ) and the scores of the other indices. Block Design, Matrix Reasoning, and Picture Concepts were the strongest subtests, while Comprehension, Digit Span, Letter-Number Sequencing, and Coding were the weakest. In the group of individuals with ASD without comorbid intellectual disability, the VCI score was higher than the scores of the other indices, and the PSI had the lowest score. Vocabulary and Similarities were the strongest subtests, and Digit Span and Coding were the weakest. In the TD group, there were no significant differences between the indices, and Picture Concepts and Vocabulary were the strongest subtests. Nader et al. [11] compared the Wechsler cognitive profile of 25 ASD people aged 6 to 16 to that of 22 TD people. In the ASD group, the PRI received significantly the highest score, and in the TD group, the WMI received the lowest score significantly.

Although this is a strong overselling of the power of IQ score, the WISC-IV provides useful information about the effect of people’s culture, biology, maturation, and differences in interventions on cognitive functioning [6]. All of the cited literature involves research conducted in Western countries, and the effect of culture on this issue has not been seriously taken into consideration, so it is difficult to determine whether the behavior and cognition of people with ASD vary among different cultures [12]. Intercultural studies allow researchers to recognize the common and distinctive characteristics of the disorder between different cultures, and provide hypotheses about the cognitive nature of developmental disorders. No studies have yet directly compared the phenotypes of ASD children from different cultural backgrounds [13]. One of the aims of this study is to provide information that can help compare different communities.

Assessing the cognitive profile in the ASD group poses the question of whether there is a relationship between ASD symptoms and the cognitive profile. Most studies on the possibility of a relationship between ASD symptoms and performance on cognitive tests have been conducted using previous versions of the WISC or other intelligence assessment tests. Oliveras-Rentas et al. [9] investigated the relationship between ASD symptoms and the cognitive profile using the Autism Diagnostic Observation Schedule (ADOS) [14] and the WISC-IV, and concluded that there was a negative relationship between the scores of the Communication subtest in the ADOS, and the VCI and the PSI and their subtests in the WISC-IV. There was also a negative relationship between the Vocabulary and Comprehension subtests of the WISC-IV and the scores of the Reciprocal Social Interaction subtest of the ADOS.

Due to the reported 37–85% prevalence of simultaneous attention deficit hyperactivity disorder (ADHD) and ASD symptoms [15], investigating the relationship between ADHD symptoms and the cognitive profile of ASD people is important. Measuring the cognitive profile of those with ADHD using the WISC-IV showed weaknesses in the WMI and PSI [16,17], but Oliveras-Rentas et al. did not confirm the significance of this relationship in the ASD group with ADHD symptoms [9]. Another aim of the present study was to evaluate the cognitive profiles of people with ASD and ADHD symptoms using the WISC-IV in a developing society.

### Aims

The general aim of this study was to evaluate the cognitive profile in Iranian people with high-functioning ASD and compare it to the profile of TD people, and then investigate the relationship between the cognitive profile in those with high-functioning ASD with ASD and ADHD symptoms through three sub-aims:(1)Investigating the cognitive profile of those with ASD and TD using the Persian version of the WISC-IV, and comparing it to previous research findings;(2)Investigating the possible relationship between ASD symptoms and performance on the WISC-IV;(3)Investigating the relationship between ADHD symptoms and the cognitive profile of people with high-functioning ASD.

These goals offer data for comparing different communities from varying cultural backgrounds. Thus, the relationship between the WISC-IV’s cognitive profile and ASD and ADHD symptoms can be assessed for the first time in a developing country.

## 2. Materials and Methods 

### 2.1. Participants

Two groups of people participated in this study, those with high-functioning ASD (*n* = 30), and the typical development or TD group (*n* = 30).

#### 2.1.1. Participants with High-Functioning ASD

The participants of this study consisted of 27 boys (90%) and 3 girls (10%) with high-functioning ASD, aged 6 to 16 (mean = 11 years and 1 month, SD = 2 years and 9 months). In terms of family income, 14 participants had a high socioeconomic status, seven had a moderate status, and nine had a low socioeconomic status. A total of 13 participants had parents with associate degrees or lower, and 17 had parents with bachelor’s degrees or higher. All of the participants were Iranian and lived in Tehran, Iran. 

The samples were selected through convenience sampling. A total of 121 children and adolescents aged 6 to 16 with a diagnosis of high-functioning ASD were referred by the Iranian Special Education Organization, the Autism Charity, Roozbeh Psychiatric Hospital, the specialized clinics of schools of rehabilitation sciences, and 10 specialized clinics for children with autism in Tehran, Iran. All of the participants had previously been evaluated by psychiatrists, psychologists, and speech therapists. A trained evaluator from the Special Education Organization had administered the Autism Diagnostic Interview-Revised (ADI-R) to 51 (42.14%) of the parents of participants. A total of 15 families (12.39%) were unwilling to participate, and eight (6.61%) of the children were outside of the study’s age range. To review the inclusion criteria, the children’s medical records were studied at a meeting with their families and the following items were assessed: comorbid conditions such as metabolic disorders and genetic syndromes, history of neurological diseases (such as trauma, brain lesions, tumors, stroke, epilepsy, and Tourette’s syndrome), and other medical issues that could affect cognition, comorbid mental disorders such as schizophrenia or bipolar disorder, and special visual, hearing, or motor problems that could affect their performance on the test. If any of these conditions were present, the child was excluded from the study. At this stage, eight (6.61%) children were excluded. To ensure the presence of ASD symptoms at the time of the study and estimate their severity, the second version of the Gilliam Autism Rating Scale (GARS-2) was conducted by a trained person. The participants who obtained the cut-off score entered the next stage. At this stage, three (2.48%) children did not obtain the minimum cut-off score of GARS-2 and were excluded.

At the next step, a well-trained and experienced psychologist, with certification for implementing the Wechsler test in ASD people and who is currently active in the implementation of the test, conducted it in a quiet room with proper light and temperature. This person was not informed about the aims and outcomes of the study. The participants entered the study if they had an FSIQ over 70. At this stage, 57 (47.1%) children were excluded and 30 were enrolled. The parents completed the Conners’ Parent Rating Scale-Revised (Short) (CPRS-RS). Table 1 show the GARS-2 and CPRS-RS scores for this group.

#### 2.1.2. TD Participants

The TD participants were selected through cluster sampling. Some of the municipal districts of Tehran were randomly selected, and an all-boys’ and an all-girls’ school were randomly selected from the primary and lower secondary schools of each district. The TD students were randomly selected from these schools in proportion to the ASD students’ genders and dates of birth. The Strengths and Difficulties Questionnaire (SDQ), which will be described in the Research Tools section, was completed for the students by their teacher. If the subjects’ mental health was approved by the questionnaire, their physical and mental health status was reconfirmed by reviewing their educational records and talking to their teachers. The students with a personal or family history of neurological, psychiatric, or other conditions affecting brain development were excluded. Of the 30 students examined, three were replaced with three others for these reasons. Ultimately, 30 TD students were selected in proportion to the ASD participants. None of these students had any academic problems.

### 2.2. Research Tools

*The Persian version of the Wechsler Intelligence Scale-Fourth Edition (WISC-IV):* The present study used the Persian version of the WISC-IV, which has had its reliability and validity assessed in 872 Iranian children aged 6 to 16 [18]. The Introduction section discussed the components of this scale.

*The Gilliam Autism Rating Scale-Second Edition (GARS-2):* The GARS-2 is a behavior scale for use with people at ages 3 to 22. The scale includes 42 items in three subscales, including stereotyped behaviors, communication, and social interaction. The GARS-2 contains an additional 14 items that examine children’s development in the first three years of their life. These items are answered with Yes and No, and provide additional information. This study used a version of the test normalized by Samadi and McConky [19] for use among Iranians.

*The Strengths and Difficulties Questionnaire (SDQ):* The SDQ is a short screening tool that is used to determine behavioral and emotional problems in children and adolescents, and assesses five main subgroups of psychiatric symptoms, including conduct problems, hyperactivity/inattention (HI), emotional symptoms, peer relationships, and pro-social behavior. The total score of the first four yields the overall score in terms of problems. The impact score is also calculated to determine the impact of the children’s problems on their own and their family’s daily life [20]. The reliability and validity of the Persian version of this questionnaire have been calculated in two separate studies [20,21].

*Conners’ Parent Rating Scale-Revised (Short) (CPRS-RS):* This short 27 item form is suitable for ages 3 to 17. The parents use this scale to score their child’s behavior over the prior month on a four-point scale. The scale has four subscales, including (1) oppositional, (2) cognitive problems/inattention, (3) hyperactivity, and (4) the ADHD index [22]. In Iran, the standardization and reliability of the CPRS-R have been assessed in two separate studies [23,24]. The present study used the CPRS-RS to measure the ADHD index along with three other scales, and its goal was not to make a diagnosis. Corbett et al. [25] also recommended this tool for classifying ASD people with ADHD comorbidity.

*The self-administered demographic and economic questionnaire:* This questionnaire contains questions about the children’s demographics along with parental data, including their employment status, annual income, and level of education.

### 2.3. Statistical Analysis

The data were statistically analyzed with SPSS software using both descriptive and analytical measures. The mean and standard deviation were used for the descriptive part of the analysis. For the analytical part, the Kolmogorov–Smirnov test was used to ensure the normal distribution of the data. The multivariate analysis of variance (MANOVA) was used to compare the existing differences in the Full-Scale Intelligence Quotient (FSIQ) and the Wechsler intelligence indices, and subtests between the ASD and TD groups. Repeated measure ANOVAs were used to investigate within-group differences among the index and subtest scores. Eta-squared (ŋ^2^) was reported as the effect size in ANOVA. The standard difference is the difference between the two test means, divided by the square root of the pooled variance [26]. Pearson’s correlation coefficient was used to examine the relationship between the FSIQ, the Wechsler intelligence indices and subtests, and the GARS-2 and CPRS-RS subtests. The level of statistical significance was set at P < 0.05. The mean of each index or subtest in the groups was used to draw the group analysis plots of the data.

### 2.4. Ethics

This study received an ethics code (ir.iums.rec.1394.9211363204) from the Ethics Committee of Iran University of Medical Sciences. Written consent was obtained from the parents for their children’s participation in the study, and verbal consent was obtained from the participants themselves.

## 3. Results

### 3.1. Intergroup Comparison

The two groups were not significantly different in terms of age and gender, but they were in terms of the FSIQ. The presence of significant FSIQ differences among the groups presents a methodological dilemma. Some authors have argued that FSIQ should not be included as a covariate because psychiatric disorders may directly cause mild FSIQ deficits compared to individuals without psychiatric disorders and that controlling for the FSIQ removes a portion of the variance that is associated specifically with psychiatric disorders. In contrast, some authors have argued that the FSIQ should be controlled [27]. Because this issue has not been resolved definitely, the results are reported both with and without controlling for the FSIQ between two groups. To match the FSIQ for the reasons mentioned, the researchers did not use the FSIQ as a covariate in statistical analysis, but, instead, the participants with an average FSIQ were selected from the high-functioning ASD (*n* = 13) and TD (*n* = 13) groups. The statistical tests showed no significant differences in the FSIQ scores between the two groups.

The comparison of the WISC-IV index and subtest scores between the high-functioning ASD and TD groups without controlling for the FSIQ showed a significant difference on combined dependent variable of WISC-IV subtest scores (F (10, 49) = 14.30, P = 0.001, Wilk’s Lambada = 0.25, Partial ŋ2 = 0.74) and index scores (F (10, 48) = 3.08, P = 0.005, Wilk’s Lambada = 0.60, Partial ŋ2 = 0.39). Analysis of each of the dependent variables, as shown in Table 2, revealed that participants with ASD were weaker than TD participants in all WISC-IV index and subtest scores. The mean difference produced maximum effect sizes for the VCI and Picture Concept subtest, and minimum effect sizes for the WMI and Digit Span subtest. After matching two groups in term of FSIQ, the effect of the groups were still statistically significant on combined dependent variables of WISC-IV subtest scores (F (10, 15) = 3.02, P = 0.02, Wilk’s Lambada = 0.33, Partial ŋ2 = 0.66), but not in index score. Analysis of each of the dependent variables, as shown in Table 2, revealed that participants with ASD were weaker significantly than TD participants in the subtests of Picture Concepts (F (1, 24) = 19.48, P = 0.001, Partial ŋ2 = 0.45), Comprehension (F (1, 24) = 12.90, P = 0.001, Partial ŋ2 = 0.35), Vocabulary (F (1, 247) = 6.30, P = 0.01, Partial ŋ2 = 0.20), and Symbol Search (F (1, 24) = 4.32, P = 0.04, Partial ŋ2 = 0.15).

### 3.2. Intragroup Comparison

The cognitive profile of those with high-functioning ASD showed a significant difference between the different indices of intelligence and the FSIQ (F (4, 116) = 3.92, P = 0.005, ŋ2 = 0.119). The post hoc tests revealed the mean scores of the FSIQ (mean = 85.63, SD = 12.51) and WMI (mean = 85.2, SD = 15.43) to be at the same level, and to be significantly lower than the PSI (mean = 93, SD = 14.49) and PRI scores (mean = 93.2, SD = 11.87).

There was a significant difference in the TD group with varying profiles between the different intelligence indices and the FSIQ score (F (4, 116) = 27.96, P = 0.001, ŋ2 = 0.491). The post hoc tests revealed the mean WMI (mean = 99.56, SD = 9.78) to be significantly lower than the FSIQ score and all of the intelligence indices. The FSIQ (mean = 111.87, SD = 9.62) and PRI (mean = 113.23, SD = 11.14) scores were at a similar level, and were significantly lower than the VCI (mean = 118.27, SD = 10.99) score. The PSI scores (mean = 115.43, SD = 9.71) were significantly higher than the FSIQ scores. Table 3 presents the p-values for the comparison of the indices between the ASD and TD groups.

The analysis of the intelligence subtests in the ASD group showed a significant difference between the subtests (F (9, 261) = 9.90, P = 0.001, ŋ2 = 0.255). The post hoc tests revealed the most statistically significant high scores in the Matrix Reasoning subtest (mean = 10.76, SD = 2.67), while the Comprehension (mean = 5.03, SD = 3.46) and Picture Concepts subtests (mean = 5.66, SD = 2.30) were at the same level and had significantly lower scores than the other subtests, with the exception of the Letter–Number Sequencing subtest (mean = 6.10, SD = 2.72). The subtests Block Design (mean = 8.93, SD = 3.70), Vocabulary (mean = 8.43, SD = 4.44), Similarities (mean = 8.20, SD = 4.02), and Symbol Search (mean = 7.86, SD = 2.37) had significantly higher scores than Letter–Number Sequencing.

There was also a significant difference between the subtests in the TD group (F (9, 261) = 20.69, P = 0.001, ŋ2 = 0.416). The post hoc tests revealed that the Vocabulary subtest (mean = 15.83, SD = 1.93) had the highest mean scores by a significant margin and the Digit Span (mean = 9.46, SD = 2.44) and Letter–Number Sequencing subtests (mean = 9.30, SD = 2.33) had the most statistically significant low mean scores. The subtests of Symbol Search (mean = 12.33, SD = 2.23) and Matrix Reasoning (mean = 12.73, SD = 2.65) had significantly higher scores than the Coding (mean = 11.20, SD = 2.23) and Picture Concepts subtests (mean = 10.86, SD = 1.79). Table 4 presents the p-value for the comparison of the subtests between the two groups.

Assessing the correlation between the intelligence indices and the subtests of the GARS-2 and CPRS-RS using Pearson’s correlation coefficient showed a significant negative correlation between the Communication subtest of the GARS-2 and the VCI, and a positive correlation between the Social Interaction subtest of the GARS-2 and the WMI. There was also a significant negative correlation between the cognitive problems/inattention subtest of the CPRS-RS and the VCI and WMI. The FSIQ showed a significant negative correlation with the Communication subtest of the GARS-2 and the cognitive problems/inattention subtest of the CPRS-RS.

Assessing the correlation between the intelligence subtests and the GARS-2 and CPRS-RS subtests using Pearson’s correlation coefficient showed a significant negative correlation between the Communication subtest of the GARS-2 and the Vocabulary and Comprehension subtests. The Social Interaction subtest of the GARS-2 also had a significant positive correlation with the Digit Span and Picture Concepts subtests. The cognitive problems/inattention index of the CPRS-RS had a significant negative correlation with the Vocabulary, Comprehension, and Digit Span subtests. In addition, the hyperactivity and ADHD index subscales of the CPRS-RS had a significant negative correlation with the Comprehension subtest of the WISC-IV (Table 5).

### 3.3. Individual Comparison

The profiles of the ASD and TD groups were plotted according to the mean scores (one graph per group, *n* = 2). Concomitantly, mean intelligence indices were plotted for every subject in both groups (*n* = 60). To show dispersion from the general group pattern, the profile of each subject was then compared to the profile of his or her own group. This indicated that the profiles of 15 subjects (50%) in the ASD group and 21 subjects (70%) in the TD group were similar to the profiles obtained from their own groups (profiles of the indices and the subtests of intelligence in the ASD and TD groups are shown in Figure 1a,b). Remarkably, the profile of seven subjects (23%) in the TD group was similar to the profile of the ASD group, but the profile of none of the people in the ASD group was similar to that in TD group. A graph of the individual and group subtests was also plotted. The results showed that the dispersion in the profiles of the subtests in those in both groups was so high that no one had a similar profile to the profile in his own group or the other group.

The individual analysis showed that 50% of the subjects in the ASD group were average in terms of the PRI and PSI. The WMI and VCI scores were below average in 70.1% and 56.7% of the subjects, respectively (according to the group analysis scores, the mean PRI and PSI scores were average in the ASD group, and the WMI and VCI scores were below average).

The individual analysis of the TD subjects showed that the VCI, PRI, and PSI scores were above average in 70%, 56.7%, and 73.4% of the subjects, respectively, while the WMI was average and below in 86.7% of them (according to the group analysis of the TD people, all of the indices were above average except for the WMI, which was average). In other words, more than 50% of the subjects in both groups showed a similar group level of analysis in terms of the intelligence indices.

## 4. Discussion

### 4.1. Intergroup Comparison

The comparison of the two groups showed a significant difference in all of the indices and subtests of the WISC-IV. Although it was not possible to match all the participants of the two groups in terms of their FSIQ, the comparison of the results of 13 ASD subjects were matched for FSIQ with 13 TD individuals showed a difference between the two groups in the VCI and subtests of Picture Concepts, Comprehension, Vocabulary, and Symbol Search. Although this small sample size was not representative of all of the differences, those significant differences between the two groups, which are discernible despite the small sample population, indicate a large difference in the two groups.

### 4.2. Intragroup Comparison

Based on the findings of this study on the cognitive profile of the people with high-functioning ASD, the mean PRI and PSI were significantly higher than the mean WMI and FSIQ. A number of studies have shown the higher competence of the PRI compared to the other indices [8,10,11]. This index was higher than the WMI in a study by Oliveras-Rentas et al. [9], but, in contrast to previous studies, the PSI was higher than the WMI and the FSIQ in both groups in this study. Weaknesses have been reported in the PSI in people with high-functioning ASD in studies conducted using the WISC-IV [7,8,9,10] and WISC-III [28]. This study does not clarify why the subjects’ processing speeds were higher than their FSIQ and WMI scores in both groups, but, considering the impact of culture on people’s performance in intelligence tests [29,30], background differences may be one of the reasons. 

In line with previous research findings, the WISC-IV subtests’ analysis showed a good competence in Matrix Reasoning [7,8,9,10,11] and weaknesses in Comprehension [7,9,10]. Unlike in Wechsler’s preliminary study [7], or the studies by Mayes and Calhoun [8] and Nader, Jelenic, and Soulieres [10], the Picture Concepts subtest was one of the weakest subtests in the present study. In other words, in this study, of the two motor-free and untimed subtests of the PRI, Matrix Reasoning received the highest total score and Picture Concepts the lowest. The Matrix Reasoning subtest is relatively culture-free, and its high scores indicate good processing of visual information and nonverbal abstract reasoning skills. The weaknesses of the ASD group in the Picture Concepts subtest might represent a weakness in nonverbal concept forming and rigid thought processes. The ability of ASD people to engage in abstract reasoning might create innovative and unconventional relationships between pictures [6], and might have thus reduced the Picture Concepts score.

In this study, the PSI and VCI scores were higher than the FSIQ, the VCI score was higher than the PRI score, and the WMI score was lower than the scores of all of the indices in the TD group. A study by Nader [11] found no statistically significant differences between the index scores of TD people, while in a study by Nader, Jelenic, and Soulieres [10], the WMI score was significantly lower than the FSIQ, PRI, and VCI scores. In the analysis of the subtests, the Vocabulary subtest had the highest score and Digit Span and Letter-Number subtests had the lowest scores, just as in a study by Nader et al. [10].

This study found a negative correlation between the Communication subtest of the GARS-2, the FSIQ, and the VCI, Vocabulary, and Comprehension subtests of the WISC-IV. That is, the more the parents reported communication problems, the weaker the children scored on the FSIQ, VCI, and Vocabulary and Comprehension subtests. Two other studies also found a negative correlation between verbal skills in the WISC and communication problems in the ADOS [9,31]. There was also a negative correlation between communication problems in the GARS-2 and FSIQ in the present study, while the FSIQ had no significant relationship with the Social Interaction score in the GARS-2. Kenworthy et al. [32] also concluded that the FSIQ can predict more scores in the field of communication and everyday skills, but there are no relationships between the FSIQ and social skills. Other studies have also suggested the impossibility of using the FSIQ for predicting adaptive behaviors, especially social skills, in people with high-functioning ASD [33,34,35]. According to Klein et al. [36], social skills have been severely impaired in people with high-functioning ASD, and their scores cannot be predicted based on the FSIQ.

According to the results, there is a significant positive correlation between the Social Interaction subscale of the GARS-2 and the WMI and the Picture Concepts and Digit Span subtests, which raises the question of why a cognitive measure such as the WMI and the subtests of Picture Concepts and Digit Span have a positive correlation with the Social Interaction subscale of the GARS. According to a study by Joseph, Tiger-Flusberg, and Lord [37], children with higher nonverbal skills obtain significantly lower scores in the social function of the ADOS, regardless of their overall abilities and verbal skills. Another study in 2009 showed that, although verbal skills indicate better performance in the ASD group, the difference between verbal and nonverbal intelligence is more related to the manifestation of social interactions [31]. Based on the comparison of the results of this and previous studies, the emergence of such patterns is not unexpected.

Just as in the study by Oliveras-Rentas et al. [9], there was no significant relationship between the ADHD index of the CPRS-RS and the indices of the WISC-IV in the ASD people, while there was a significant negative correlation between the ADHD and hyperactivity indices and the Comprehension subtest of the WISC-IV. Bruce et al [38] found a negative correlation between verbal Comprehension and ADHD symptoms in ADHD people. The cognitive problems/inattention index of the CPRS-RS showed a significant negative correlation with the FSIQ and the Vocabulary, Comprehension, and Digit Span subtests of the WISC-IV in the ASD group. Naglieri et al. [39] found no relationship between the cognitive problems/inattention index of Conners’ Parent Rating Scales-Revised (Long-Form) and the subtests of the WISC-III, but a significant relationship was observed between cognitive problems/inattention in the Conners’ Teachers Rating Scales-Revised (Long-Form) and the FSIQ and the VCI and freedom from the distractibility subtests of the WISC-III. 

### 4.3. Individual Comparison 

In the analysis of the pattern of the IQ indices, 21% of the subjects in the TD group showed the same pattern as the subjects in the ASD group. This finding probably confirms the results of studies conducted to assess the features of ASD in TD people [40], but in the present study, there was no TD group profile in the ASD subjects. Perhaps plotting profiles might serve as an assistive factor in the final clinical decision for ruling out ASD. In other words, perhaps ASD might be ruled out in a subject who is an ASD suspect and has a profile of the TD subjects.

## 5. Limitation

The findings of the present study should be considered with respect to its limitations. Findings might be different using a larger sample size with a FSIQ matched TD control group at baseline. One of the limitations was that lack of prior research and findings on the topic from other developing countries made it impossible to lay a foundation for understanding the research problem we were investigating. We propose similar studies be done by other researchers to understand the possibility that these cognitive profile differences might be attributable to ADHD or other learning differences, rather than ASD?

## 6. Conclusions

This study examined the cognitive profile of people with high-functioning ASD and compared it to that of those with TD, and examined the relationship between this profile and ASD and ADHD symptoms. The findings can help compare the results between Eastern and Western societies. There were many similarities between the results of this research and previous studies; however, a high processing speed in both the ASD and TD groups was the distinguishing point between this study and similar studies, which may be attributed to cultural factors. Further studies with larger sample sizes are required for examining this hypothesis. According to the results of this study, it may be possible to use drawing plots of the intelligence indices as an assistive factor in the final clinical decision for ruling out the chance of ASD.

## Figures and Tables

**Figure 1 behavsci-09-00020-f001:**
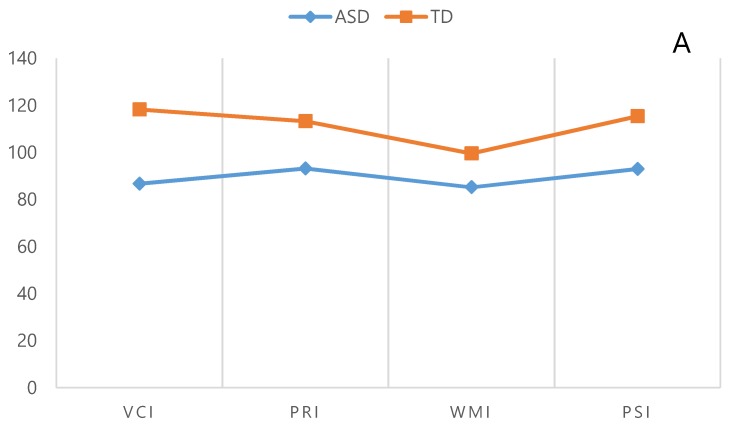
Profiles of the indices and subtests of the WISC-IV in the ASD and TD groups are shown. (**A**) Profiles of the indices of the WISC-IV in the ASD and TD groups. (**B**) Profiles of the subtests of the WISC-IV in the ASD and TD groups.

**Table 1 behavsci-09-00020-t001:** The mean (SD) of the indices and the subtests of the GARS-2 and CPRS-RS in the ASD sample (N = 30).

Scales		Mean (SD)	95% CI
GARS-2	GARS Autism Index	74.18 (11.85)	[69.73, 78.59]
	Stereotyped Behaviors—SS	6.1 (2.66)	[5.10, 7.09]
	Stereotyped Behaviors—%ile	15.46 (15.25)	[9.77, 21.16]
	Communication—SS	4.26 (1.38)	[3.74, 4.78]
	Communication—%ile	6.8 (3.41)	[5.52, 8.07]
	Social interaction-SS	6.1 (1.56)	[5.51, 6.68]
	Social interaction—%ile	11.7 (5.84)	[9.51, 13.88]
	Total Standard Score	16.46 (4.42)	[14.81, 18.11]
	Rank Percent	23.13 (15.59)	[17.31, 28.95]
CPRS-R:S	ADHD Index	60.83 (7.28)	[58.14, 63.55]
	Oppositional	54.97 (9.70)	[51.34, 58.59]
	Cognitive Problems/ Inattention	60.40 (9.03)	[57.03, 63.77]
	Hyperactivity	64.23 (11.59)	[59.91, 68.56]

CI: Confidence Interval; SS: Standard Score; %ile: Percentage Score; GARS-2: Gilliam autism rating scale-2; CPRS-R:S: Conners’ Parent Rating Scale-Revised, (Short); ADHD: Attention-deficit/hyperactivity disorder.

**Table 2 behavsci-09-00020-t002:** Comparison of the mean (SD) of the indices and the subtests of the WISC-4 between the high-functioning ASD and TD groups with FSIQ matched and not matched (ASD (*n* = 13) and TD (*n* = 13)), and without (ASD (*n* = 30) and TD (*n* = 30)) FSIQ matched.

Indices and the Subtests of the WISC-4	FSIQ	ASD		TD		Intergroup Comparisons	
		Mean (SD)	95% CI	Mean (SD)	95% CI	P-Value	Cohens d
VCI	Not Matched	86.70 (19.18)	[79.53, 93.86]	118.27 (10.99)	[114.16, 122.37]	0.001	2.01
	Matched	100.31 (14.98)	[91.25, 109.36]	111.38 (8.71)	[106.11, 116.65]	0.030	0.90
Similarities	Not Matched	8.20 (4.02)	[6.69, 9.70]	11.96 (3.59)	[10.62, 13.30]	0.001	0.98
	Matched	9.92 (4.36)	[7.28, 12.56]	9.53 (2.84)	[7.81, 11.25]	P > 0.05	-
Vocabulary	Not Matched	8.43 (4.44)	[6.77, 10.09]	15.83b(1.93)	[15.11, 16.55]	0.001	2.16
	Matched	12.15 (3.13)	[10.26, 14.04]	14.84 (2.26)	[13.47, 16.21]	0.019	0.98
Comprehension	Not Matched	5.03 (3.46)	[3.73, 6.32]	11.26 (3.00)	[10.14, 12.38]	0.001	2.23
	Matched	7.00 (3.26)	[5.02, 8.97]	10.69 (1.75)	[9.63, 11.75]	0.001	1.41
PRI	Not Matched	93.20 (11.87)	[88.76, 97.63]	113.23 (11.14)	[109.07, 117.39]	0.001	1.74
	Matched	100.23 (11.30)	[93.39, 107.06]	103.54 (7.25)	[99.15, 107.92]	P > 0.05	-
Block Design	Not Matched	8.93 (3.70)	[7.55, 10.31]	12.10 (3.26)	[10.88, 13.31]	0.001	0.90
	Matched	9.61 (3.61)	[7.42, 11.80]	9.46 (2.22)	[8.11, 10.80]	P > 0.05	-
Picture Concepts	Not Matched	5.66 (2.30)	[4.80, 6.52]	10.86 (1.79)	[10.19, 11.53]	0.001	2.52
	Matched	7.15 (1.51)	[6.23, 8.07]	10.00 (1.73)	[8.95, 11.04]	0.001	1.75
Matrix Reasoning	Not Matched	10.76 (2.67)	[9.76, 11.76]	12.73 (2.65)	[11.74, 13.72]	0.006	0.74
	Matched	12.15 (2.64)	[10.55, 13.74]	11.23 (2.27)	[9.85, 12.60]	P > 0.05	-
WMI	Not Matched	85.20 (15.43)	[79.43, 90.96]	99.56 (9.78)	[95.91, 103.21]	0.001	1.11
	Matched	93.61 (15.58)	[84.20, 103.03]	95.61 (7.77)	[90.91, 100.31]	P > 0.05	-
Digit Span	Not Matched	7.60 (4.06)	[6.08, 9.11]	9.46 (2.44)	[8.55, 10.37]	0.035	0.55
	Matched	9.76 (3.44)	[7.68, 11.85]	8.92 (2.39)	[7.47, 10.37]	P > 0.05	-
Letter–Number Sequencing	Not Matched	6.10 (2.72)	[5.08, 7.11]	9.30 (2.33)	[8.42, 10.17]	0.001	1.25
	Matched	6.92 (3.30)	[4.92, 8.91]	8.46 (2.02)	[7.23, 9.68]	P > 0.05	-
PSI	Not Matched	93.00 (14.49)	[87.58, 98.41]	115.43 (9.71)	[11.80, 119.06]	0.001	1.81
	Matched	102.62 (13.89)	[94.21, 11.01]	108.08 (5.88)	[104.52, 11.63]	P > 0.05	-
Coding	Not Matched	7.53 (3.49)	[6.22, 8.83]	11.20 (2.23)	[10.36, 12.03]	0.001	1.25
	Matched	9.53 (3.59)	[7.36, 11.71]	10.07 (1.75)	[9.01, 11.13]	P > 0.05	-
Symbol Search	Not Matched	7.86 (2.37)	[6.98, 8.75]	12.33 (2.23)	[11.49, 13.16]	0.001	1.94
	Matched	9.53 (3.59)	[7.36, 11.71]	10.76 (1.42)	[9.90, 11.62]	0.048	0.45
FSIG	Not Matched	85.63 (12.51)	[80.96, 90.31]	11.87 (9.62)	[108.27, 115,46]	0.001	2.35
	Matched	97.08 (9.06)	[91.60, 102.55]	102.85 (5.04)	[99.80, 105.90]	P > 0.05	-
Age (months)	Not Matched	133.17 (33.12)	[120.80, 14.53]	134.43 (32.06)	[122.46, 146.41]	0.881	-
	Matched	146.62 (30.69)	[122.46, 151.39]	127.23 (39.06)	[103.62, 150.84]	P > 0.05	-

CI: Confidence Interval, ASD: Autism Spectrum Disorder, TD: Typical Development, FSIQ: Full-Scale Intelligence quotient, VCI: Verbal Comprehension Index, PRI: Perceptual Reasoning Index, WMI: Working Memory Index, PSI: Processing Speed Index.

**Table 3 behavsci-09-00020-t003:** P-value of the repeated measures analysis for the comparison of the WISC-IV indices between the ASD (*n* = 30) and TD groups (*n* = 30).

Index	Group	FSIQ	VCI	PRI	WMI	PSI
FSIQ	ASD					
	TD					
VCI	ASD	0.633				
	TD	0.001				
PRI	ASD	0.001	0.068			
	TD	0.236	0.022			
WMI	ASD	0.839	0.682	0.006		
	TD	0.001	0.001	0.001		
PSI	ASD	0.003	0.119	0.943	0.018	
	TD	0.013	0.247	0.191	0.001	

FSIQ: Full-Scale IQ, VCI: Verbal Comprehension Index, PRI: Perceptual Reasoning Index, WMI: Working Memory Index, PSI: Processing Speed Index, ASD: Autism Spectrum Disorder, TD: Typical Development.

**Table 4 behavsci-09-00020-t004:** P-value of the repeated measures analysis for the comparison of the WISC-IV subtest score between the ASD (*n* = 30) and TD groups (*n* = 30).

Subtest	Group	Similarities	Vocabulary	Comprehension	Block Design	Picture Concepts	Matrix Reasoning	Digit Span	Letter–Number Sequencing	Coding	Symbol Search
Similarities	ASD										
	TD										
Vocabulary	ASD	0.734									
	TD	0.001									
Comprehension	ASD	0.001	0.001								
	TD	0.507	0.001								
Block Design	ASD	0.434	0.611	0.001							
	TD	0.828	0.001	0.456							
Picture Concepts	ASD	0.004	0.001	0.314	0.001						
	TD	0.111	0.001	0.148	0.061						
Matrix Reasoning	ASD	0.005	0.005	0.001	0.006	0.001					
	TD	0.250	0.001	0.069	0.279	0.001					
Digit Span	ASD	0.478	0.372	0.007	0.113	0.009	0.001				
	TD	0.004	0.001	0.003	0.001	0.015	0.001				
Letter–Number Sequencing	ASD	0.005	0.013	0.157	0.003	0.447	0.001	0.061			
	TD	0.001	0.001	0.001	0.001	0.005	0.001	0.780			
Coding	ASD	0.515	0.322	0.007	0.139	0.004	0.001	0.940	0.064		
	TD	0.260	0.001	0.475	0.094	0.473	0.005	0.004	0.002		
Symbol Search	ASD	0.660	0.474	0.001	0.128	0.001	0.001	0.708	0.006	0.533	
	TD	0.603	0.001	0.192	0.667	0.003	0.475	0.001	0.001	0.029	

ASD: Autism Spectrum Disorder, TD: Typical Development.

**Table 5 behavsci-09-00020-t005:** Pearson’s correlations between the WISC-IV IQ index and the subtest score and ASD and ADHD symptomatology.

Scale		WISC-IV Index and Subtests														
		Similarities	Vocabulary	Comprehension	VCI	Block Design	Picture Concepts	Matrix Reasoning	PRI	Digit Span	Letter–Number Sequencing	WMI	Coding	Symbol Search	PSI	FSIQ
**GARS-2**	**GARS Autism Index**	−0.154	−0.238	−0.320	−0.273	−0.235	0.078	0.010	−0.096	0.243	0.089	0.218	−0.027	−0.044	0.071	−0.118
	**Stereotyped Behaviors**	−0.191	−0.126	−0.347	−0.252	−0.181	0.045	0.047	−0.057	0.242	0.079	0.209	−0.045	0.062	0.132	−0.131
	**Communication**	−0.189	−0.466 **	−0.417 *	−0.410*	−0.151	−0.230	−0.196	−0.267	−0.072	0.020	−0.041	0.039	−0.345	−0.204	−0.378 *
	**Social interaction**	0.057	−0.071	0.031	−0.001	−0.148	0.363 *	0.113	0.109	0.436 *	0.144	0.393 *	−0.020	0.141	0.136	0.224
	**Total Standard Score**	−0.154	−0.247	−0.329	−0.281	−0.208	0.083	0.007	−0.080	0.277	−0.105	0.252	−0.022	0.056	0.063	−0.114
	**Rank Percent**	−0.149	−0.233	−0.316	−0.264	−0.254	0.053	−0.016	−0.127	0.196	0.076	0.177	−0.032	0.036	0.078	−0.129
**CPRS-R:S**	**Oppositional**	−0.146	−0.065	−0.191	−0.155	−0.112	0.044	−0.047	−0.071	−0.003	−0.084	−0.044	0.118	0.158	0.215	0.014
	**Cognitive Problems/Inattention**	−0.345	−0.390 *	−0.404 *	−0.423 *	−0.150	−0.229	−0.334	−0.344	−0.526 **	−0.295	−0.531 **	0.161	−0.108	−0.068	−0.434 *
	**Hyperactivity**	−0.171	−0.147	−0.415 *	−0.282	−0.126	−0.015	−0.025	−0.089	−0.089	−0.108	−0.128	0.131	0.134	0.230	−0.097
	**ADHD Index**	−0.264	−0.162	−0.383 *	−0.296	−0.156	−0.038	−0.196	−0.213	−0.241	−0.192	−0.262	0.331	−0.054	0.207	−0.178

GARS-2: Gilliam Autism Rating Scale-2, CPRS-R:S: Conners’ Parent Rating Scale-Revised (Short), VCI: Verbal Comprehension Index, PRI: Perceptual Reasoning Index, WMI: Working Memory Index, PSI: Processing Speed Index, FSIQ: Full Scale IQ, ADHD: Attention-deficit/hyperactivity disorder, p < 0.05*, p < 0.01**.

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
