# Peer review of "The Cognitive Profile of People with High-Functioning Autism Spectrum Disorders"

_behavsci, 2019, doi:10.3390/bs9020020_

Reviewer 1 Report

In this research paper, the authors analysed the cognitive profiles in high-functioning children with autism.

The authors made a great effort, the study is well designed and conducted.

As minor issues:

- it would be interesting to add in the introduction a short description on autism incidence or prevalence in Iran, to better understand the validity of the population studied in this paper.

- In conclusion, add more on the outcomes of the study. Why its significance will be helpful in clinical settings?

Author Response

My co-authors and I would like to sincerely thank the Reviewers for the comments that were provided upon review of our manuscript. To the best of our abilities, we believe we have addressed all the comments. Our responses to the Review comments are below in red color.

Reviewer #1: Comments

Point 1: it would be interesting to add in the introduction a short description on autism incidence or prevalence in Iran, to better understand the validity of the population studied in this paper.

Response 1: Prevalence rate of ASD in Iran was reported.

Point 2: In conclusion, add more on the outcomes of the study. Why its significance will be helpful in clinical settings?

Response 2: In the clinical setting, this outcome can be used as a contributing factor in the final clinical decision and judgement for ruling out the chance of ASD which has been points out in the conclusion section as follows:

According to the results of this study, it may be possible to use drawing plots of the intelligence indices as an assistive factor in the final clinical decision for ruling out the chance of ASD”.

Reviewer 2 Report

Thank you for asking me to review this manuscript entitled “The Cognitive Profile of People with High-Functioning 3 Autism Spectrum Disorders” submitted for review in the journal Behavioral Sciences.  The authors present a study of cognitive profiles on the WISC-IV of a group of individuals with autism spectrum disorder and neurotypical peers between the ages of 6 and 16 years.  I have a number of suggested edits for the authors.

Introduction, Lines 33-35:  The authors assert: “The Diagnostic and Statistical Manual of Mental Disorders, Fifth Edition (DSM-5) defines autism 34 spectrum disorders (ASD) as neurodevelopmental disorders that are associated with intellectual disability frequently.”  I would like to see a statistic of comorbidity of ASD and ID.  How much is “frequently”?  According to the last published prevalence study of the US Center for Disease Control and Prevention (CDC) – the reported a rate of co-morbidity of ASD and ID of approximately 30% - that does not strike me as “frequently”.

Introduction, Lines 35-36: The authors make the statement that a high IQ is what determines the use of the terminology of “high functioning”.  I am puzzled by the use of “high functioning” immediately following the authors citation of the DSM-5.  This might lead the readers to erroneously assume that the DSM-5 uses this nomenclature.  Which is inaccurate.  The DSM-5 does not use IQ scores to classify individuals with ASD.  The propose using the individual’s estimated level of support needs.  In fact, DSM-5 does not use “low functioning” or “high functioning” ASD anywhere.   I would suggest that the authors rephrase the next sentence as well (line 36-37): “Knowing the intelligence profile of people with ASD is necessary for interpreting their diagnosis.”  The assessment of the individual’s intellectual functioning is not an essential part of the diagnostic process of ASD.  It might have value in characterizing a sample or developing educational strategies or planning interventions but it has nothing to do with a diagnosis of ASD – at least according to the DSM-5, which the authors cite at the very beginning of this section.

I suggest dropping the use of “Asperger syndrome” and refer to these individuals as having ASD without comorbid intellectual disability – consistent with DSM-5 proposed terminology.

I would contest the authors statement on p. 2, lines 71-72: “The WISC-IV provides useful information about the effect of people’s culture, biology, maturation, and differences in interventions on cognitive functioning.”  This is a strong oversell of the powers of IQ scores.  I think the authors should calibrate their assertion of the implications of knowing a person’s intellectual ability.  I am not sure how the authors could support the assertion that IQ scores inform about a person’s culture.  This is a very controversial and inflammatory assertion.  I suggest rephrasing this sentence and toning it down some.

It is not clear to me how the diagnosis of ASD was made.  The authors reported that the individuals were evaluated by an interdisciplinary team but do not state that they made a formal diagnosis of ASD.  The authors mention using the ADI-R but it is unclear how this was done since they report “… administered the Autism Diagnostic Interview-Revised (ADI-R) to 51 (42.14%) of the participants.” But the ADI-R is a standardized clinical interview that is conducted with the subject’s parents/guardians/knowledgeable caregiver and not the participant him/herself as the manuscript seems to indicate. Finally, the authors report using the GARS as their primary tool to establish the diagnosis of ASD, when the GARS is a “screening” tool at best.  It is not mentioned clearly that the ASD diagnosis was made by a trained clinician or interdisciplinary team. Not indicating clearly how the authors established “caseness” is a serious methodological error.

 p. 4, lines 159-160: The authors should correct a statement made regarding the GARS. The GARS is not a “diagnostic tool” for ASD.  It is more clearly a “screening tool” that assists in ruling-in and ruling-out ASD but not a definitive diagnosis. 

I would like to see more detail on how the “comparison” group was selected.  There is a statement indicating that they had not academic problems – but does this mean that they were NOT selected among other students receiving special education services?  The procedures section seems to indicate that ALL the ASD students had been referred to the Special Education services.  Why not?  Is this not a limitation – how do the groups compare?  Why are both groups not described in the same detail? I see more details about the ASD group and fewer details about the comparison TD group.  Was the TD group assessed on the measures of ADHD? 

Table 2 – title header seems to be incomplete. Please review.

Discussion, line 68: I suggest replacing “complained about” with “reported”.

I would like to see some discussion about the possibility that these cognitive profile differences might be attributable to ADHD or other learning differences – rather than ASD? 

Another suggestion might be to tone down a little this comment “Since the present study was considered to be a groundbreaking study in its field” made in the Limitations section.  Although this study has contributed to our knowledge and understanding of cognitive profiles, this study remains largely a replication of previous methodology and findings.  I would suggest that the authors include a limitation regarding sample size and representation of sample – can this be generalized to all people with ASD and TD?

Author Response

My co-authors and I would like to sincerely thank the Reviewers for the comments that were provided upon review of our manuscript. To the best of our abilities, we believe we have addressed all the comments. Our responses to the Review comments are below in red color.

Reviewer #2: Comments

Point 1: Lines 33-35:  The authors assert: “The Diagnostic and Statistical Manual of Mental Disorders, Fifth Edition (DSM-5) defines autism 34 spectrum disorders (ASD) as neurodevelopmental disorders that are associated with intellectual disability frequently.”  I would like to see a statistic of comorbidity of ASD and ID.  How much is “frequently”?  According to the last published prevalence study of the US Center for Disease Control and Prevention (CDC) – the reported a rate of co-morbidity of ASD and ID of approximately 30% - that does not strike me as “frequently”.

Response 1: The rate of co-morbidity of ASD and ID were both reported.

Point 2: Introduction, Lines 35-36: The authors make the statement that a high IQ is what determines the use of the terminology of “high functioning”.  I am puzzled by the use of “high functioning” immediately following the authors citation of the DSM-5.  This might lead the readers to erroneously assume that the DSM-5 uses this nomenclature.  Which is inaccurate.  The DSM-5 does not use IQ scores to classify individuals with ASD.  The propose using the individual’s estimated level of support needs.  In fact, DSM-5 does not use “low functioning” or “high functioning” ASD anywhere.  I would suggest that the authors rephrase the next sentence as well (line 36-37): “Knowing the intelligence profile of people with ASD is necessary for interpreting their diagnosis.”  The assessment of the individual’s intellectual functioning is not an essential part of the diagnostic process of ASD.  It might have value in characterizing a sample or developing educational strategies or planning interventions but it has nothing to do with a diagnosis of ASD – at least according to the DSM-5, which the authors cite at the very beginning of this section.

Response 2: These sentences were rewritten.

Point 3: I suggest dropping the use of “Asperger syndrome” and refer to these individuals as having ASD without comorbid intellectual disability – consistent with DSM-5 proposed terminology.

Response 3: Change was done.

Point 4: I would contest the authors statement on p. 2, lines 71-72: “The WISC-IV provides useful information about the effect of people’s culture, biology, maturation, and differences in interventions on cognitive functioning.”  This is a strong oversell of the powers of IQ scores.  I think the authors should calibrate their assertion of the implications of knowing a person’s intellectual ability.  I am not sure how the authors could support the assertion that IQ scores inform about a person’s culture.  This is a very controversial and inflammatory assertion.  I suggest rephrasing this sentence and toning it down some.

Response 4: we do agree that this assertion which indicates IQ scores have relation with a person culture is a very strong justification but this assertion has been supported by a very outstanding reference (This sentence is from Handbook of psychological assessment (written by Groth-Marnat G.) in -chapter 5 (Wechsler Intelligence Scales) - Section (Testing of Intelligence: Pros and Cons) but we have rephrased this sentence in the shade of your useful comment as follow:

Although this is strong overselling of the power of IQ score but ….

Point 5: It is not clear to me how the diagnosis of ASD was made.  The authors reported that the individuals were evaluated by an interdisciplinary team but do not state that they made a formal diagnosis of ASD. 

Response 5: The samples were recruited from several different sectors. Although we strongly agree that ADI-R   is a golden standard instrument for the diagnosis of ASD (albeit in its moderate and sever forms), ADI-R was already administered for 51 (42.14%) of the participants and the other sectors use different diagnostic tools. Therefore we tried to apply a single scale to re-evaluate all of the sample members at the beginning. Therefore based on the first authors training we picked GARS-2 considering its Iranian norms.  

Point 6: The authors mention using the ADI-R but it is unclear how this was done since they report “… administered the Autism Diagnostic Interview-Revised (ADI-R) to 51 (42.14%) of the participants.” But the ADI-R is a standardized clinical interview that is conducted with the subject’s parents/guardians/knowledgeable caregiver and not the participant him/herself as the manuscript seems to indicate.

Response 6: This sentence was rewritten.

Point 7: Finally, the authors report using the GARS as their primary tool to establish the diagnosis of ASD, when the GARS is a “screening” tool at best.  It is not mentioned clearly that the ASD diagnosis was made by a trained clinician or interdisciplinary team. Not indicating clearly how the authors established “caseness” is a serious methodological error.

Response 7: Base on (Samadi SA, McConkey R. The utility of the Gilliam autism rating scale for identifying Iranian children with autism. Disability and rehabilitation. 2014 Mar 1;36(6):452-6), which is approved by James E Gilliam the scale developer, this scale although is a second level screening tool but if it is   used by a trained professional along with clinical observation of the subject and with reviewing his/her developmental profile it can be considered as a diagnostic tool. In this research, we used GARS-2 to ensure the presence of ASD symptoms at the time of the study based on the Autism Index and estimate their severity based on their percentage rate, the second version of the Gilliam Autism Rating Scale (GARS-2) was conducted by a trained person(The first author is officially trained as a GARS-2 administrator). It also worth to stress that GARS2 is used to understand the Autism Index of the sample at the beginning of the study.

Point 8: p. 4, lines 159-160: The authors should correct a statement made regarding the GARS. The GARS is not a “diagnostic tool” for ASD.  It is more clearly a “screening tool” that assists in ruling-in and ruling-out ASD but not a definitive diagnosis.

Response 8: This sentence was rewritten.

Point 9: I would like to see more detail on how the “comparison” group was selected.  There is a statement indicating that they had not academic problems – but does this mean that they were NOT selected among other students receiving special education services?  The procedures section seems to indicate that ALL the ASD students had been referred to the Special Education services.  Why not?  Is this not a limitation – how do the groups compare?  Why are both groups not described in the same detail? I see more details about the ASD group and fewer details about the comparison TD group.  Was the TD group assessed on the measures of ADHD?

Response9: The TD group was recruited from a public school in which the student receives no special education. Base on Iranian regulation all the students who are able to pass the school entry screening program are permitted to be registered in public school. and, children who fail to pass screening and diagnostic evaluation program (in Iran it is called “Sanjesh” program and is mandatory for all the 5 years old children who want to be registered at the first grade of the primary school) are entitled to receive special services (Samadi SA, Mahmoodizadeh A, McConkey R. A national study of the prevalence of autism among five-year-old children in Iran. Autism. 2012 Jan;16(1):5-14.).

For ASD groups, family were also visited therefore, we could collect more data compared to the TD group which we had no similar chance.

For the TD group SDQ was administered to understand the presence of ADHD symptoms.

“The SDQ is a short screening tool that is used to determine behavioral and emotional problems in children and adolescents and assesses five main subgroups of psychiatric symptoms, including conduct problems, hyperactivity/inattention (HI), emotional symptoms, peer relationships, and pro-social behavior”

Point 10: Table 2 – title header seems to be incomplete. Please review.

Response 10: Title header was completed.

Point 11: Discussion, line 68: I suggest replacing “complained about” with “reported”.

Response 11: the phrase was changed

Point 12: I would like to see some discussion about the possibility that these cognitive profile differences might be attributable to ADHD or other learning differences – rather than ASD?

Response 12: Thank you for your interesting suggestion. We have added the following suggestion to the limitation section.

Point 13: Another suggestion might be to tone down a little this comment “Since the present study was considered to be a groundbreaking study in its field” made in the Limitations section.  Although this study has contributed to our knowledge and understanding of cognitive profiles, this study remains largely a replication of previous methodology and findings.  I would suggest that the authors include a limitation regarding sample size and representation of sample – can this be generalized to all people with ASD and TD?

Response 13: Thank you for your suggestion.   This sentence was rewritten.
